# Methylsulfonylmethane Improves Knee Quality of Life in Participants with Mild Knee Pain: A Randomized, Double-Blind, Placebo-Controlled Trial

**DOI:** 10.3390/nu15132995

**Published:** 2023-06-30

**Authors:** Akifumi Toguchi, Naoto Noguchi, Toshihiro Kanno, Akira Yamada

**Affiliations:** 1Department of Research and Development, Chlorella Industry Co., Ltd., Hisatomi 1343, Chikugo 833-0056, Fukuoka, Japan; 2Research Center for Innovative Cancer Therapy, Kurume University, Kurume 830-0011, Fukuoka, Japan

**Keywords:** methylsulfonylmethane, knee pain, JKOM

## Abstract

Methylsulfonylmethane (MSM) is a food ingredient present in small amounts in many foods, and its anti-inflammatory effects have been reported. We conducted a randomized, double-blind, placebo-controlled trial of oral consumption of MSM on mild pain of the knee joint in healthy Japanese participants. A total of 88 participants were enrolled in this study and randomly assigned to MSM consumption (*n* = 44) and placebo control (*n* = 44) groups. Both groups of participants took 10 tablets, each containing 200 mg MSM or lactose, per day for 12 weeks. The primary outcome of this study was measured values of the total score of the Japanese Knee Osteoarthritis Measure (JKOM) at 12 weeks after the test sample consumption. Safety evaluation was performed through physical examination, urine analysis, peripheral blood test, and medical interview. The total scores at 12 weeks in the MSM and placebo groups as the primary outcome were significantly different (*p* = 0.046). The health condition of JKOM also improved after MSM consumption (*p* = 0.032). The questionnaire results also suggested improvement in the knee and systemic health. This study indicated that MSM oral consumption improved both knee and systemic health conditions in healthy participants who experienced mild pain in the knee joint.

## 1. Introduction

Sulfur composes 34% of methylsulfonylmethane (MSM), a food ingredient [1]. Sulfur is naturally present in small amounts in many foods, such as milk, vegetables, and fruits [1]. Sulfur is the fourth most abundant mineral element after calcium, phosphorus, and potassium and is present in relatively large amounts in hair, nails, skin, and cartilage. MSM is considered to be a source of sulfur in these tissues, and an animal study involving rats confirmed the incorporation of sulfur derived from ingested MSM into these tissues [1,2].

Discomfort in joints is caused by local destruction and inflammation of the joint(s), and the acute phase of discomfort often becomes a chronic state and may progress to more severe diseases, such as osteoarthritis (OA) and rheumatoid arthritis. To prevent the progression of joint problems, daily dietary care is essential. Thus, many dietary components that maintain a normal healthy state of joints have been investigated [3]. In particular, OA is a painful disease caused by abnormal cartilage homeostasis and increased cartilage destruction that impairs the synthesis of new chondrocytes [4]. A further imbalance between cartilage synthesis and degradation can lead to inflammation and exacerbation of OA [5]. Demographic joint replacement may be performed if the symptoms become severe; however, most patients remain on symptomatic therapy [6]. Therefore, researchers are focusing on the prevention and early detection of OA [7].

MSM has been suggested as a supplement for the prevention of OA due to its pleiotropic effects that help in joint protection. MSM has been shown to promote osteoblast differentiation via the Jak2/STAT5b pathway [8] and regulate the expression of RUNX2, an upstream signal of SP7 (osterix) that controls the expression of osteogenic genes, such as SPARC (osteonectin), SSP1 (osteopontin), COL1A1-COL1A2 (collagen type 1), and BGLAP (osteocalcin) [9]. The administration of MSM has been shown to promote chondrogenesis in zebrafish. The promotion of osteogenic differentiation by MSM through the induction of bone morphogenetic protein 2 (BMP-2) [10] or transglutaminase-2 (TG2) has been demonstrated in an in vitro study [11]. The anti-inflammatory effect of MSM through the inhibition of the transcriptional activity of nuclear factor-kappa B (NF-kB) has been reported [12,13]. The NF-kB pathway is considered to be an amplification system of inflammation-related cytokines, chemokines, and adhesion molecules [14]. Thus, inhibition of NF-kB by MSM causes suppression of interleukin (IL)-1, IL-6, and tumor necrosis factor-α (TNF-α) expression at the mRNA level [15].

The chondrogenic and anti-inflammatory effects of MSM and the incorporation of the ingested MSM-derived sulfur into the cartilage suggest an application of MSM for the prevention or treatment of inflammation-related joint diseases. In animal models, therapeutic effects of MSM on experimental arthritis have been reported, i.e., the score of arthritis in the models was improved in the MSM treatment groups [16,17]. Human clinical studies of MSM in patients with OA have also been conducted [18,19,20,21]. Administration of MSM relieved joint pain and reflected the improvement of several indicators of joint conditions, such as the Visual Analog scale (VAS), Western Ontario and McMaster University Arthritis score (WOMAC), 36-item Short Form survey (SF-36), and Lequensne index [18,19,20,21]. In addition, improvements in the stiffness and swelling of joints have been observed [19]. Control of the local inflammation at the joints by MSM administration not only affected the joint conditions but also improved systemic physical functions, i.e., physical function-related indicators of the WOMAC, SF-36, and Aggregated Locomotor Function scores in patients with OA were improved in the MSM groups [18,19,20,21]. Both clinical and pre-clinical animal studies have supported the therapeutic usefulness of MSM for joint diseases such as OA; however, there is no direct evidence to support prophylactic use for joint diseases or use to maintain the normal condition of joints in a healthy population. In the present study, we conducted a randomized, double-blind, placebo-controlled trial of oral consumption of MSM on mild pain of the knee joint in healthy participants.

## 2. Materials and Methods

### 2.1. Study Aim

This study aimed to determine the effect of oral consumption of MSM on knee joint quality of life (QOL) in mild knee pain participants.

### 2.2. Participants

Healthy Japanese participants with mild pain of the knee joint(s) were recruited from 18 May 2021 to 21 August 2021, by Ortho Medico (Tokyo, Japan), a contracted research organization, using a website (https://www.go106.jp/). Participants were eligible for inclusion if they met all the inclusion criteria and did not meet the exclusion criteria. The inclusion criteria were as follows: (1) Japanese, (2) participants aged 20 years or older, (3) healthy participants, and (4) participants who are experiencing mild pain in the knee joint(s). The exclusion criteria were as follows: (1) participants who have at least one previous instance of malignant tumor in medical history, heart failure, or myocardial infarction; (2) participants who have a pacemaker or an implantable cardioverter defibrillator (ICD); (3) participants who are undergoing medical treatment of knee disease, such as knee OA (KOA); (4) participants who are currently under the treatment for cardiac arrhythmia, hepatic disorder, renal disorder, cerebrovascular disorder, rheumatism, diabetes mellitus, dyslipidemia, hypertension, or other chronic diseases; (5) participants who take “Foods for Specified Health Uses”, “Foods with Functional Claims”, or other functional food/beverage in daily use; (6) participants currently taking or using medicines (including herbal medicines, cold patch, and topical medication) and supplements; (7) participants who are allergic to medicines and/or the test food related products; (8) participants who are pregnant, breast-feeding, or planning to become pregnant; (9) participants who have been enrolled in other clinical trials within the last 28 days before the agreement to participate in this trial or plan to participate another trial during this trial; and (10) participants who are determined to be ineligible to participate in the study by the physician. Additional participant selection criteria for the screening of participants were as follows: (1) participants who are determined to be eligible to participate in the study by the physician, (2) participants who are identified as being the Kellgren–Lawrence grade (K-L grade) [22] either 0 or 1 on X-ray at screening. Participants who met criteria 1 and 2 were assigned in order, starting with those with the highest total score of the Japanese Knee Osteoarthritis Measure (JKOM) [23].

### 2.3. Study Design

This study was a randomized, double-blind, placebo-controlled trial. Participants were randomly assigned to two groups, namely, an MSM consumption group and a placebo control group, with a 1:1 ratio. The details of this study were explained, and written informed consent was obtained from all participants. Both groups of participants took 10 tablets per day, five tablets each at breakfast and dinner for 12 weeks. The dominant ingredient of the tablets of the MSM consumption group was 200 mg MSM per tablet and that of the placebo control group was the same amount of lactose. Lactose was adopted as a placebo, referring to a clinical trial of supplement intake for knee pain [24]. Single doses of lactose up to 12 g have been reported to be acceptable [25]. OptiMSM^®^ (Bergstrom Nutrition, Vancouver, WA, USA), a high purity and quality product of MSM made by the distillation method [26], was used as the active substance of the tablets. The test tablets were provided by CIC Fronter (Tokyo, Japan). The MSM and placebo tablets were manufactured under Dietary Supplement Good Manufacturing Practice. MSM and placebo were identical in appearance and were indistinguishable by smell. Participants visited the Medical Corporation Seishinkai Takara Clinic (Tokyo, Japan) four times (weeks 0, 4, 8, and 12) between 11 September 2021 and 25 December 2021. The intervention was conducted from 11 September 2021 to 25 December 2021.

This study was conducted in accordance with the principles of the Declaration of Helsinki. The trial protocol and any amendments were approved by the ethics committee of the Takara Clinic, Medical Corporation Seishinkai (approval number: 2204-00145-0004-0E-TC), and statistical analysis was performed by Ortho Medico. This trial has been registered in the UMIN Clinical Trials Registry (UMIN000044258).

### 2.4. Outcomes

The measured values of the total score of JKOM at 12 weeks after the test sample consumption were considered the primary outcome of this study. The secondary outcomes were as follows: (1) the measured values of the total score of JKOM at 4 and 8 weeks after consumption; (2) the differences in the total score of JKOM between screening (0 weeks) and 4, 8, or 12 weeks; (3) the measured values at 4, 8, and 12 weeks and amounts of changes between screening (0 weeks) and 4, 8, or 12 weeks after assessment of VAS [27], scores of pain and stiffness in knees, conditions in daily life, general activities, and health conditions of JKOM, Japanese Orthopedic Association (JOA) score [28], and scores of pain on walking, pain on ascending or descending stairs, range of motion, and joint effusion of JOA. Plasma levels of IL-1 beta, IL-6, high sensitivity C-reactive protein (Hs-CRP), procollagen II C-Terminal propeptide (PIICP), and collagen Type II cleavage (C2C); and (4) the values of the individual items of the JKOM questionnaire and the JOA score at 4, 8, and 12 weeks. Due to the outbreak of COVID-19 infection, the originally planned measurement of collagen Type 1 and 2 cleavage marker was not available and thereby the measurement marker was changed to C2C. Safety evaluation was performed through physical examination, urine analysis, peripheral blood test, and medical interview.

The JKOM questionnaire was written in Japanese and an English translation had been published [23]. JKOM is a patient-based self-assessment that includes five related items that reflect the Japanese cultural lifestyle. JKOM questionnaire assesses subjective knee symptoms by participants answering a VAS about the degree of knee pain and 25 questions. The four items consist of 8 questions about “Ⅱ: pain and stiffness conditions”, 10 questions about “Ⅲ: condition in daily life”, 5 questions about “Ⅳ: general activities” and 2 questions about “Ⅴ: health conditions” for a total of 25 questions. Participants answered each question on a 5-point Likert scale ranging from no disability (0 points) to severe disability (4 points) for chronic symptoms in the past few days and health status in the past month. The Ⅱ to Ⅴ items were summed, giving total scores from 0 to 100. Total scores and scores in each item were compared between MSM and placebo groups. The concurrent and construct validity of the JKOM has been validated against the WOMAC Index scale and SF-36 [23], JOA scores and diagnoses several knee conditions, with higher scores indicating less joint damage [28]. Walking pain (0–30), stair climbing pain (0–25), range of motion (0–35), and joint fluid (swelling; 0–10) are quantified by the physician. This gives a total score range of 0–100.

Diets three days before each test date were surveyed. The diet study used the Calorie and Nutrition Diary (CAND), developed by Ortho Medico [29]. A recorded dietary questionnaire was brought and submitted at each visit.

### 2.5. Sample Size and Statistical Analysis

The target sample size was 80, assuming an ineligibility rate of approximately 10%. The sample size was calculated with the following assumptions: Cohen’s d = 0.80, type I error rate = 0.05, power = 80%, and the ratio of the two groups = 1:1. All participants, physicians, investigators, and analysts were blinded to the allocation information. The staff of Ortho Medico provided computer-generated random numbers using Statlight#11 Ver. 2.10 software (Yukms, Kawasaki, Japan). Data were described as means and SDs, median and minimum to maximum, or 95% confidence interval. The Chi-square test was used to compare sex and K-L grade between the two groups. Mean differences between two individual groups were analyzed by Welch’s *t*-test or Mann–Whitney U test according to the normality of the data. The linear mixed model (LMM) was applied to analyze the time course of the effect [30]. LMM was modeled using groups, weeks (time), and their interactions as fixed effects. The significance of differences between groups was investigated by intergroup interaction and time. LMM was used to assess the difference in the rate of change between pre- and post-ingestion of MSM and placebo. The significance of Analyses of primary and secondary outcomes was based on the per-protocol analysis set (PPS) and that of the safety evaluation was based on the safety analysis population. Statistical analyses were performed using SPSS Statistics version 23 (IMB, Armonk, NY, USA). The statistical analysis plans were finalized before the key opening.

## 3. Results

### 3.1. Participant Flow

From 25 May 2021 to 21 August 2021, 163 healthy Japanese participants with mild pain of the knee joint(s) were screened for enrollment (Figure 1). A total of 88 participants were enrolled and randomly assigned to the MSM consumption (*n* = 44) and placebo control (*n* = 44) groups. One participant in the MSM consumption group withdrew from the trial during the study period because there were no visits to the clinic after 8 weeks of consumption. Due to low compliance with test food consumption (less than 80%), 5 and 3 participants in the MSM and control groups, respectively, were excluded from the analysis. Finally, data from 80 and 88 participants were analyzed for efficacy and safety, respectively, to compare MSM and placebo controls in this study. Baseline demographics in the MSM and placebo groups were balanced, and there was no significant difference between the two groups at week 0 (Table 1). The mean compliance rates of test sample consumption in the PPS of MSM and placebo control groups were 100.4% and 100.5%, respectively.

### 3.2. JKOM Scores, JOA Scores, and Inflammation Markers

The measured values of JKOM scores and JOA scores are shown in Table 2. The JKOM total scores at 12 weeks after the test sample consumption of MSM and placebo control groups as primary outcome were 8.1 and 10.9 points, respectively, with the difference being significant (difference = −2.8, *p* = 0.046, 95%CI: −5.5, −0.1). These scores were not significantly different from the baseline scores of each group (18.2 and 19.9, respectively). Health condition scores at 12 weeks in the MSM group were significantly lower than those in the placebo group (difference = −0.6, *p* = 0.027, 95%CI: −1.0, −0.1), indicating improvement. JKOM of MSM and placebo groups at 12 weeks was lower than the baseline of each group, i.e., differences in the scores were −1.3 and −0.9, respectively, and the difference between groups was −0.5 (*p* = 0.032, 95%, CI: −0.9, −0.0) (Appendix A). There were no significant between-group differences in the amount of change before and after other JKOM items. Significant differences in the JOA total scores among the different time periods of the two groups were not observed (Table 2). There was also no significant difference in the amount of change in JOA score between the MSM and placebo groups during any week of comparison (Appendix A).

The results of inflammatory markers and type II collagen biomarkers are shown in Table 3. Among the type II collagen biomarkers, PIICP at 8 weeks in the MSM group showed a transient upward trend. Statistical analysis in a linear mixed model of the change in PIICP from baseline to 8 weeks showed a significant increase in the change in MSM, 23.6 in the MSM group and 11.3 in the placebo group (difference = 30, *p* = 0.008, 95% CI: 7.8, 52.2) (Table 4). However, PIICP values at 12 weeks in both the MSM and placebo groups decreased from baseline. Other markers in the blood did not differ between the two groups (Table 3). There was also no significant difference in the amount of change in IL-1 beta, IL-6, and Hs-CRP between the MSM and placebo groups during any week of comparison (Appendix A). A decrease in JKOM total scores and health condition scores in the MSM group suggested improvement of knee conditions and systemic health conditions.

### 3.3. Questionaries

The JKOM questionnaire consists of a total of 25 items: Q1 to Q8 for pain and stiffness in knees, Q9 to Q18 for conditions in daily life, Q19 to Q23 for general activities, and Q24 to Q25 for health conditions. Answers for each question were chosen from a scale of 0 to 4: 0 for no pain at all or good condition, and 4 for most severe or bad condition. Only four items with significant changes in scores were observed during the study period, and the results are shown in Figure 2. The score changes were observed for the following items: is Q2, morning pain; Q3, nocturnal pain; Q8, pain while standing; Q24, health condition. Morning pain, pain while standing, and health condition were significantly reduced at 12 weeks in the MSM group compared with those in the placebo group. Nocturnal pain and pain while standing were also reduced after 8 weeks of MSM consumption. There were no significant changes in the scores regarding the other items of the JKOM questionnaire. Dietary intake surveys in the three days before measurement using CAND showed wide variation by nutrient and no factors associated with the results of this study. 

### 3.4. Safety Evaluation

The results of physical examination, urine analysis, peripheral blood test, and medical interview revealed no health problems during the study period, and no adverse events were reported (Appendix A).

## 4. Discussion

This is the first clinical trial of MSM oral consumption in healthy participants who felt mild pain in the knee joint rather than patients with OA. The JKOM total scores at 12 weeks after test sample consumption in the MSM group were significantly lower than those in the placebo group. The health conditions of the JKOM of the MSM group at 12 weeks were also significantly lower than those of the placebo group. Decreases in the JKOM total scores and the health condition scores indicated improvement of the knee and systemic health conditions of the participants by MSM consumption. The MSM group showed significant improvements compared to placebo at 12 weeks, in terms of morning pain, pain while standing, and health condition in the JKOM questionnaires. These results indicate that MSM alleviates wake-up pain and standing pain and improves general health and quality of life in healthy individuals with knee mild pain in daily life.

OA is common in the hands, knees, hips, and spine and manifests as inflammation, stiffness, and loss of range of motion [31]. Knee pain is a common symptom in patients with KOA [32]. KOA is one of the leading diseases affecting older people worldwide, causing long-term disability and reduced quality of life [33,34]. Early disease progression of OA results in disruption of chondrocyte metabolism and increased secretion of degradative enzymes, such as collagenase [4]. Subsequently, synoviocytes ingest cartilage degradation products via phagocytosis and induce inflammation through the production of inflammatory cytokines. KOA progression is diagnosed by classifying it into K-L grades based on X-ray findings [22]. In this study, MSM consumption improved the JKOM total score in participants with knee mild pain and a low KL grade. In addition, it was found that MSM consumption improved the responses to pain-related questionnaires. Improvements in pain-related questionnaires by MSM consumption in OA patients, but not in healthy participants, have been shown previously [18,19,20]. Most cases of OA are associated with inflammatory or cartilage-bone turnover markers [35]. The PIICP, a cartilage metabolism marker, was improved at 8 weeks in the MSM group. PIICP and C2C, which are cartilage synthesis and degradation markers, respectively, are used in combination for clinical assessment of KOA progression [36,37]. In this study, alteration of C2CP and C2C by MSM consumption was not observed; thus, the effect of MSM consumption on cartilage metabolism is limited, although MSM has been reported to improve cartilage metabolism in animal models and in vitro studies [9]. He et al. reported that urine levels of C2C reflected the changes in cartilage conditions more rapidly than those in serum levels [38]. Therefore, if we used urine samples instead of serum samples, MSM consumption might affect the marker levels. Further studies are needed to confirm this.

The JKOM score is a questionnaire on the QOL of KOA based on Japanese lifestyle [23]. The JKOM “pain and stiffness” scale focuses on knee pain during daily activities such as stair use, knee bending, standing up from sitting, and walking. In JKOM scores, the pain and stiffness items from Q1 to Q8 have been shown to be strongly correlated with medial and lateral KOA knee pain [39,40]. Pain-related questions Q2, Q3, and Q8 improved in the MSM group at week 12 compared with the placebo group. MSM has also been shown to be effective in reducing pain and physical dysfunction using the WOMAC index in KOA patients [20], suggesting that MSM may tend to improve in terms of pain. Hs-CRP is associated with the degree and severity of KOA and is a known marker of pain [41,42,43,44], but the study found no difference in Hs-CRP levels between the MSM and placebo groups. Thus, rather than reducing inflammation, the direct antioxidant effects of MSM in the knee may have contributed to pain relief.

The pathophysiology of KOA includes inflammatory and degenerative processes associated with oxidative stress. Mild inflammation may be caused by aging, which may contribute to local inflammation and affect the knee joint [45,46]. Inflammation and oxidative stress are interdependent, activate cartilage signaling pathways, and alter chondrocyte tissue homeostasis in the KOA [47,48]. Inflammatory changes reduce antioxidant activity in biofluids and cartilage, increase oxidative stress levels, and affect proteins involved in knee joint structural strength [49]. Because of this, oxidative stress and inflammation show an interdependent relationship within the knee joint of KOA, and it is thought that oxidative stress and synovial inflammation act on each other to exacerbate the pathology [50]. Although the subjects in this study were not KOA, they were aware of the deterioration of the QOL of their knees, suggesting that the level of oxidative stress in the knee joints may have increased. Since MSM is known to exhibit antioxidant activity and free radical scavenging [26], it is speculated that the antioxidant activity of the MSM also improved the QOL of the knee joint in this study. Indeed, malondialdehyde (MDA), a lipid peroxide, has been reported to be decreased in the urine of patients with knee osteoarthritis after ingestion of MSM [20]. However, since this study did not assess indicators related to antioxidant activity, further studies should also examine MDA, 4-hydroxynonenal (4-HNE), F2-isoprostane, hexanoyllysine (HEL), oxidized LDL, total antioxidant capacity (TAC), superoxide dismutase (SOD), and catalase (CAT).

As for changes in serum cytokine levels, studies have investigated the effects of MSM on serum cytokines in obese individuals [51]. In this study, they expected the anti-inflammatory effect of MSM and hypothesized that it would reduce cytokine levels, but the effect of MSM on cytokines was not clear. Moreover, in a Korean study that observed cytokine levels in 110 healthy individuals, IL-1 beta ranged from 0.17 to 39.0 pg/mL and IL-6 ranged from 0.16 to 37.7 pg/mL, indicating a wide range of indicators [52]. The data in this study met the reference point and were presented within the range of healthy subjects. Therefore, it is considered that the effect of the test food could not be found for the cytokines because (1) the value was at the level of healthy subjects, (2) it was a highly variable index, and (3) it was not a factor that was adjusted before the start of the study.

In recent years, it has been suggested that there is a gender difference in the association of knee pain inflammatory cytokines [53,54]. Perruccio AV, et al. studied the effects of gender on inflammatory cytokines such as IL-6, IL-8, IL-10, IL-1 beta, and TNF-alpha in patients with OA and showed that there were sex differences between individual inflammatory markers and pain [53]. Tschon M, et al. also systematically reviewed, focusing on gender differences and the relationship between gender and OA as primary objectives, and found that women had greater access to health care, a higher prevalence of OA, and different clinical pain and inflammation and cartilage volume reduction compared to men [54]. These studies suggest that there are gender differences in the impact of diet on knee QOL and that there may be gender differences in treatment effects. In this study, intake of a test meal significantly improved knee QOL, but it is important that future studies investigate whether it is particularly effective in men or women.

Depressive symptoms are a major comorbidity in elderly OA patients. Depressive symptoms are known to be a factor associated with both knee pain and physical function, especially self-reported physical function, in KOA patients [55,56]. Importantly, depression has been suggested to be a more important factor associated with knee pain and disability than radiographic evidence of degenerative changes in the joints [57]. In this study, improvement trends were observed in the items of “pain and stiffness of knee” and “health condition” in the JKOM score. In KOA, pain and loss of health are thought to be associated with depression, and MSM may be expected to reduce the risk of depression developing from KOA by relieving pain and maintaining a sense of health.

This study had several limitations. JKOM is associated with WOMAC and SF-36 and is a scientifically valid questionnaire to assess QOL related to KOA [23]; however, it does not fully reflect the physiological state of the symptoms. Inflammation markers, such as IL-1 beta, IL-6, and Hs-CRP, were not affected by MSM consumption, although MSM has been reported to have anti-inflammatory effects [12,13,15], IL-1 beta, IL-6, and Hs-CRP have been reported to correlate with medical conditions only in KOA of K-L grade ≥ 2 [58,59,60,61,62]. In the present study, subjects were healthy individuals who experienced knee problems with low K-L grade and exhibited low blood levels of inflammatory markers. Therefore, we need to search for more appropriate markers to assess the anti-inflammatory effects of MSM consumption in healthy individuals with knee problems.

## 5. Conclusions

This study indicated that MSM oral consumption improved knee conditions and systemic health conditions in healthy participants who were experiencing mild pain in the knee joint.

## Figures and Tables

**Figure 1 nutrients-15-02995-f001:**
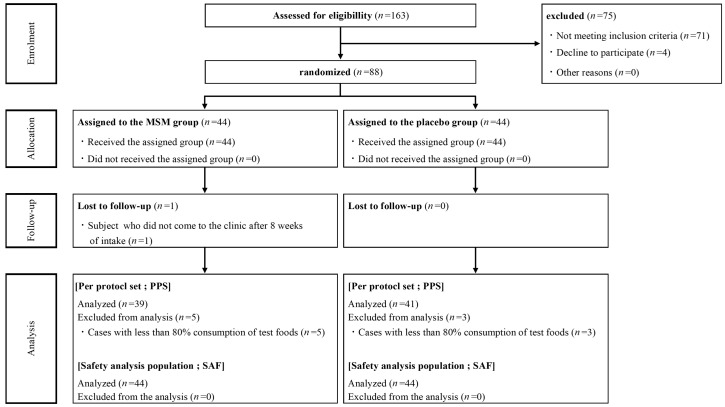
Consort flow diagram.

**Figure 2 nutrients-15-02995-f002:**
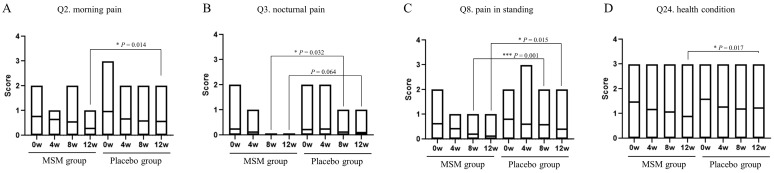
Effect of MSM consumption on JKOM questions. (**A**–**D**) The scores of each question item of the JKOM in the MSM and placebo groups in the indicated period are shown. *p*-values show comparisons between groups using the Mann–Whitney U test. * *p* < 0.05 and *** *p* < 0.001 between the MSM and placebo groups.

**Table 1 nutrients-15-02995-t001:** Background of the study participants.

(Unit)		MSM Group	Placebo Group	*p* Value ^1^	MSM Group	Placebo Group	*p* Value ^1^
		(*n* = 39)	(*n* = 41)		(*n* = 44)	(*n* = 44)	
Sex	female/male	28/11	30/11	1.000	31/13	32/12	1.000
K-L grade	grade 1/grade 0	14/25	17/24	0.652	18/26	17/27	1.000
Age	Mean ± SD	48.9 ± 13.7	48.9 ± 13.8	0.988	49.3 ± 13.7	49.2 ± 13.7	0.975
(years)	Med	52.0	48.0		52.0	47.5	
	Min–Max	26–85	23–80		26–85	23–80	
	Mean ± SD	161.5 ± 7.5	163.1 ± 7.7	0.335	161.9 ± 7.8	163.2 ± 7.5	0.424
Height	Med	161.00	161.00		161.00	161.00	
(cm)	Min–Max	144.0–174.0	148.5–180.0		144.0–188.0	148.5–180.0	
Weight	Mean ± SD	62.0 ± 15.8	60.7 ± 13.8	0.686	61.8 ± 15.7	60.3 ± 13.6	0.631
(kg)	Med	59.50	61.10		59.30	59.75	
	Min–Max	44.3–117.5	36.1–93.7		43.2–117.5	36.1–93.7	
BMI	Mean ± SD	23.6 ± 5.2	22.6 ± 4.3	0.354	23.4 ± 5.0	22.5 ± 4.2	0.353
(kg/cm^2^)	Med	22.40	21.40		22.60	21.40	
	Min–Max	18.4–42.6	16.4–36.6		17.5–42.6	16.4–36.6	
PBF	Mean ± SD	29.1 ± 10.0	27.9 ± 9.6	0.605	28.3 ± 9.7	27.6 ± 9.7	0.732
(%)	Med	26.90	26.30		26.80	25.85	
	Min–Max	13.4–58.8	12.7–55.5		13.4–58.8	12.7–55.5	
Systolic BP	Mean ±S D	119.9 ± 11.2	118.8 ± 16.4	0.716	121.3 ± 12.5	119.0 ± 16.7	0.480
(mmHg)	Med	117.0	112.0		117.5	112.5	
	Min–Max	103–149	90–159		103–157	90–159	
Diastolic BP	Mean ± SD	77.8 ± 7.8	76.7 ± 11.5	0.628	78.1 ± 8.9	76.8 ± 11.5	0.541
(mmHg)	Med	76.0	73.0		76.0	73.5	
	Min–Max	66–93	60–106		64–107	60–106	
Pulse	Mean ± SD	72.6 ± 11.3	76.5 ± 12.9	0.155	72.5 ± 10.8	75.8 ± 12.8	0.195
(bpm)	Med	74.0	77.0		74.0	74.5	
	Min–Max	45–103	55–112		45–103	55–112	

PPS, Per protocol set; SAF, Safety analysis population; *n*, number; SD, standard deviation; Med, median; Min, minimum; Max, maximum; BMI, Body mass index; PBF, percentage body fat; BP, Blood pressure; ^1^ Age and K-L grade were compared between groups using the Chi-square test. Other items were compared between groups using Welch’s *t*-test.

**Table 2 nutrients-15-02995-t002:** Japanese Knee Osteoarthritis Measure (JKOM) and Japanese Orthopaedic Association (JOA) scores from weeks 0 to 12.

		0 Week				4 Week				8 Week				12 Week			
		MSM Group	Placebo Group	Difference	*p* Value ^1^	MSM Group	Placebo Group	Difference	*p* Value ^1^	MSM Group	Placebo Group	Difference	*p* Value ^1^	MSM Group	Placebo Group	Difference	*p* Value ^1^
		Mean ± SD	Mean ± SD	(95% CI)		Mean ± SD	Mean ± SD	(95% CI)		Mean ± SD	Mean ± SD	(95% CI)		Mean ± SD	Mean ± SD	(95% CI)	
JKOM	Ⅰ:VAS	37.9 ± 21.9	41.3 ± 20.7	−3.4 (−12.9 to 6.1)	0.478	34.1 ± 22.2	32.9 ± 21.0	1.2 (−8.4 to 10.8)	0.801	25.3 ± 18.6	29.0 ± 22.3	−3.7 (−12.9 to 5.4)	0.416	24.4 ± 21.7	25.2 ± 22.1	−0.7 (−10.5 to 9.0)	0.881
	Ⅱ: Pain and stiffiness in knees	6.8 ± 2.5	7.1 ± 3.3	−0.4 (−1.7 to 0.9)	0.566	5.2 ± 2.8	5.7 ± 3.9	−0.5 (−2.0 to 1.0)	0.548	4.1 ± 2.7	4.8 ± 3.4	−0.7 (−2.1 to 0.7)	0.306	2.7 ± 2.2	3.9 ± 3.4	−1.2 (−2.4 to 0.1)	0.073
	Ⅲ: Condition in daily life	4.4 ± 3.4	4.7 ± 3.4	−0.3 (−1.8 to 1.2)	0.674	2.9 ± 4.4	3.4 ± 4.6	−0.5 (−2.5 to 1.5)	0.645	2.5 ± 2.3	2.8 ± 4.1	−0.3 (−1.8 to 1.1)	0.643	1.5 ± 1.8	2.1 ± 3.1	−0.6 (−1.7 to 0.5)	0.300
	Ⅳ: General activities	4.6 ± 2.1	5.4 ± 2.4	-0.8 (−1.8 to 0.2)	0.118	3.7 ± 1.6	4.0 ± 1.6	−0.3 (−1.0 to 0.5)	0.478	3.2 ± 1.5	3.6 ± 1.7	−0.4 (−1.1 to 0.3)	0.263	2.7 ± 1.5	3.1 ± 1.5	−0.5 (−1.1 to 0.2)	0.153
	Ⅴ: Health conditions	2.5 ± 0.9	2.6 ± 0.9	−0.1 (−0.5 to 0.2)	0.461	1.8 ± 1.0	2.0 ± 1.2	-0.2 (−0.7 to 0.3)	0.415	1.6 ± 1.0	1.7 ± 1.1	−0.2 (−0.6 to 0.3)	0.476	1.2 ± 1.0	1.8 ± 1.2	−0.6 (−1.0 to−0.1)	0.027 *
	Total JKOM score	18.2 ± 6.7	19.9 ± 7.5	−1.6 (−4.8 to 1.5)	0.305	13.6 ± 7.7	15.0 ± 10.1	−1.4 (−5.4 to 2.6)	0.492	11.3 ± 5.4	13.0 ± 8.4	−1.6 (−4.8 to 1.5)	0.307	8.1 ± 4.2	10.9 ± 7.6	−2.8 (−5.5 to−0.1)	0.046 *
JOA	Total score	95.9 ± 5.9	95.9 ± 5.0	0.0 (−2.4 to 2.5)	0.972	95.4 ± 5.2	96.5 ± 5.1	−1.1 (−3.4 to 1.2)	0.352	96.5 ± 4.1	96.5 ± 4.4	−0.1 (−1.9 to 1.8)	0.958	96.4 ± 5.3	96.6 ± 5.3	−0.2 (−2.6 to 2.1)	0.843

CI, confidence interval; ^1^ Welch’s *t*-test assessed differences between the MSM and placebo groups for 0, 4, 8, and 12 weeks. * *p* < 0.05 between the MSM and placebo groups.

**Table 3 nutrients-15-02995-t003:** Plasma concentrations of IL-1β, IL-6, Hs-CRP, CIICP, and C2C from 0 to 12 weeks.

		0 Week				4 Week				8 Week				12 Week			
		MSM Group	Placebo Group	Difference	*p* Value ^1^	MSM Group	Placebo Group	Difference	*p* Value ^1^	MSM Group	Placebo Group	Difference	*p* Value ^1^	MSM Group	Placebo Group	Difference	*p* Value ^1^
	(Unit)	Mean ± SD	Mean ± SD	(95% CI)		Mean ± SD	Mean ± SD	(95% CI)		Mean ± SD	Mean ± SD	(95% CI)		Mean ± SD	Mean ± SD	(95% CI)	
IL-1β	(pg/mL)	0.0 ± 0.1	0.0 ± 0.0	0.0 (0.0 to 0.1)	0.478	0.0 ± 0.1	0.0 ± 0.0	0.0 (0.0 to 0.0)	0.324	0.0 ± 0.0	0.0 ± 0.0	0.0 (0.0 to 0.0)	0.761	0.0 ± 0.1	0.0 ± 0.0	0.0 (0.0 to 0.1)	0.349
IL-6	(pg/mL)	1.0 ± 0.8	1.0 ± 0.6	0.0 (−0.3 to 0.3)	0.937	1.4 ± 1.0	1.4 ± 1.0	0.1 (−0.4 to 0.5)	0.807	1.2 ± 0.6	1.4 ± 0.7	-0.2 (−0.5 to 0.1)	0.227	1.2 ± 0.6	1.3 ± 0.8	−0.1 (−0.4 to 0.3)	0.737
Hs-CRP	(mg/dL)	0.2 ± 0.3	0.1 ± 0.2	0.0 (−0.1 to 0.1)	0.428	0.2 ± 0.3	0.2 ± 0.5	0.0 (−0.2 to 0.2)	0.856	0.1 ± 0.2	0.1 ± 0.2	0.0 (−0.1 to 0.1)	0.854	0.2 ± 0.4	0.1 ± 0.2	0.1 (−0.1 to 0.2)	0.306
PⅡCP	(ng/mL)	111.6 ± 58.6	94.9 ± 45.5	16.7 (−6.8 to 40.2)	0.160	121.9 ± 45.2	112.1 ± 33.7	9.7 (−8.1 to 27.6)	0.280	135.2 ± 76.9	106.2 ± 70.6	29.0 (−3.9 to 61.9)	0.083	61.4 ± 18.6	60.0 ± 22.8	1.5 (−7.8 to 10.7)	0.754
CⅡC	(ng/mL)	154.8 ± 30.0	146.3 ± 21.7	8.5 (−3.2 to 20.3)	0.152	164.6 ± 24.6	167.3 ± 31.6	−2.7 (−15.3 to 9.8)	0.667	129.0 ± 17.1	124.8 ± 16.2	4.1 (−3.3 to 11.6)	0.270	144.9 ± 22.8	141.3± 25.3	3.6 (−7.1 to 14.3)	0.508

CI, confidence interval; ^1^ Welch’s *t*-test assessed differences between the MSM and placebo groups for 0, 4, 8, and 12 weeks.

**Table 4 nutrients-15-02995-t004:** Comparison of the amount of change in PIICP at each week from week 0.

		0 to 4 Week				0 to 8 Week				0 to 12 Week			
		MSM Group	Placebo Group	Difference	*p* Value ^1^	MSM Group	Placebo Group	Difference	*p* Value ^1^	MSM Group	Placebo Group	Difference	*p* Value ^1^
	(Unit)	Mean ± SD	Mean ± SD	(95% CI)		Mean ± SD	Mean ± SD	(95% CI)		Mean ± SD	Mean ± SD	(95% CI)	
PⅡCP	(amount of change)	10.3 ± 58.0	17.2 ± 57.0	6.7 (−15.5 to 28.9)	0.554	23.6 ± 99.4	11.3 ± 84.8	30.0 (7.8 to 52.2)	0.008 **	−50.1 ± 52.7	-34.9 ± 46.8	−0.7 (−22.8 to 21.5)	0.953

CI, confidence interval; ^1^ Comparison between groups using a linear mixed model with baseline values as covariates, time points, groups, time and group interactions, baseline values and time point interactions, and study participants as factors. ** *p* < 0.01 between the MSM and placebo groups.

## Data Availability

The data used to support the findings of this study are available from the corresponding author upon reasonable request.

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
