# Peer review of "Methylsulfonylmethane Improves Knee Quality of Life in Participants with Mild Knee Pain: A Randomized, Double-Blind, Placebo-Controlled Trial"

_nutrients, 2023, doi:10.3390/nu15132995_

Round 1

Reviewer 1 Report

The authors conducted a randomized, double-blind, placebo-controlled trial of oral consumption of MSM on discomfort of the knee joint in healthy Japanese participants. The results indicated that MSM oral consumption improved both knee and systemic health conditions in healthy participants who experienced discomfort of the knee joint. This is an interesting work, but the quality of manuscript is not well prepared. There are some main problems as follows:

1.       Why the authors chose lactose as the control? More reasons should be provided.

2.       Line 163-177, please check whether these lines were related with this study.

3.       There were no descriptions of statistical analysis in methods part.

4.       The authors chose three time points, and just compared the differences at each points, but the changes before and after intervention were also suggested to be provided.

5.       Some values do not conform to the normal distribution, therefore the statistical analysis of each data should be described clearly.

6.       The discussion was too simple, the underlying mechanism based on the results of this study should be fully discussed.

The quality of English Language is not good enough and need to be impoved by a native English speaker.

Author Response

Thank you very much for the reviewers’ comments concerning our manuscript. We are sending herewith our revised manuscript in line with the reviewers’ suggestions. I would like to express my appreciation to the reviewers for the useful criticisms and advice on our paper. I hope that these revisions will merit acceptance for publication. If you find further misunderstandings or our interpretation insufficient, please let us know, we will make further corrections or additions. Following are point-by point replies to the reviewers’ comments.

Comment 1: Why the authors chose lactose as the control? More reasons should be provided.

Response 1: Thank you for your questions. Single doses of lactose up to 12 g have been reported to be acceptable [1]. Lactose was chosen as a placebo in this study because lactose was used as a placebo in clinical trials on taking supplements for mild knee pain [2]. Information on the quality and preparation of tablets has also been added, and we have revised Study Design section to the above content (p.3 lines 119-121 and p.3 lines124-126).

[1] Suchy FJ, Brannon PM, Carpenter TO, Fernandez JR, Gilsanz V, Gould JB, Hall K, Hui SL, Lupton J, Mennella J, Miller NJ, Osganian SK, Sellmeyer DE, Wolf MA. NIH consensus development conference statement: Lactose intolerance and health. NIH Consens State Sci Statements. 2010 Feb 24;27(2):1-27. PMID: 20186234.

[2] Braham R, Dawson B, Goodman C. The effect of glucosamine supplementation on people experiencing regular knee pain. Br J Sports Med. 2003 Feb;37(1):45-9; discussion 49. doi: 10.1136/bjsm.37.1.45. PMID: 12547742; PMCID: PMC1724589.

Comment 2: Line 163-177, please check whether these lines were related with this study.

Response 2: Thank you for pointing it out. We have removed this sentence.

Comment 3: There were no descriptions of statistical analysis in methods part.

Response 3: Thank you for your feedback. We have added statistical details to the Statistical Analysis section (p.4 lines 179-187).

Comment 4: The authors chose three time points, and just compared the differences at each points, but the changes before and after intervention were also suggested to be provided.

Response 4: Thank you for your advice. We have added the analysis results of the amount of change for each item in the supplementary tables (supplementary material). We also have added text about the amount of changes to the Results section (p.6 lines 220-222, p.6 lines 223-225 and p.6 lines 233-235).

Comment 5: Some values do not conform to the normal distribution, therefore the statistical analysis of each data should be described clearly.

Response 5: Thank you for your feedback. We have added detailed statistical analysis methods in the Statistical Analysis section. We added a note about the selection of statistical analyzes with normal distributions and non-normal distributions (p.4 lines 179-187).

Comment 6: The discussion was too simple, the underlying mechanism based on the results of this study should be fully discussed.

Response 6: According to the reviewer’s suggestion, we added the sentences to the discussion section on mechanisms related to pain (p.9 lines 300-311), antioxidants (p.9 lines 312- p.10 lines 330), inflammatory markers(p.11 lines 331-340), gender-based difference (p.11 lines 341-352) and depression risk (p.11 lines 353-361) in this study.

Reviewer 2 Report

The manuscript meets the remit of the area and would be of interest to the readership. The study aims to assess the impact of MSM on knee joint discomfort in a healthy Japanese cohort.  Ethical aspects of the study are considered, and the introduction has relevant background science.

The manuscript although generally well-written should be reviewed for typographical errors, e.g.  fruit instead of flute line 28. 

You seem to have left several paragraphs from the instructions for authors in the method line 163-177.  These should obviously be removed.

It would help to describe what variables contribute to the JKOM score in the materials and methods itself.

Was differences in diet controlled for, eg controlling for dietary sources of MSM?

Although the aim of the study is to examine prophylactic effects of MSM, the nature of knee discomfort should be defined more clearly.  Was preceding duration of discomfort or aetiology controlled for e.g. acute versus chronic underlying cause? What score was considered relatively high on the JKOM? Line 108-109?

Was the absolute decrease in JKOM score in MSM versus placebo groups from week 0 to week 12 significant. 

Data is presented clearly.

There is discussion of the results and suggestions for future studies that are valid.

The manuscript meets the remit of the area and would be of interest to the readership. The study aims to assess the impact of MSM on knee joint discomfort in a healthy Japanese cohort.  Ethical aspects of the study are considered, and the introduction has relevant background science.

The manuscript although generally well-written should be reviewed for typographical errors, e.g.  fruit instead of flute line 28. 

You seem to have left several paragraphs from the instructions for authors in the method line 163-177.  These should obviously be removed.

It would help to describe what variables contribute to the JKOM score in the materials and methods itself.

Was differences in diet controlled for, eg controlling for dietary sources of MSM?

Although the aim of the study is to examine prophylactic effects of MSM, the nature of knee discomfort should be defined more clearly.  Was preceding duration of discomfort or aetiology controlled for e.g. acute versus chronic underlying cause? What score was considered relatively high on the JKOM? Line 108-109?

Was the absolute decrease in JKOM score in MSM versus placebo groups from week 0 to week 12 significant. 

Data is presented clearly.

There is discussion of the results and suggestions for future studies that are valid.

Author Response

Thank you for your careful review.  Your comments were very helpful in improving this manuscript. If you find further misunderstandings or our interpretation insufficient, please let us know, we will make further corrections or additions. Following are point-by point replies to the reviewers’ comments.

Comment 1: The manuscript although generally well-written should be reviewed for typographical errors, e.g.  fruit instead of flute line 28. 

Response 1: Thank you for pointing it out. We checked the spelling and corrected the typos.

Comment 2: You seem to have left several paragraphs from the instructions for authors in the method line 163-177.  These should obviously be removed.

Response 2: Thank you for pointing it out. We have removed this sentence.

Comment 3: It would help to describe what variables contribute to the JKOM score in the materials and methods itself.

Response 3: Thank you for your advice. We added the detailed measurement method of JKOM score and JOA score (p.4 lines 154-168).

Comment 4: Was differences in diet controlled for, eg controlling for dietary sources of MSM?

Response 4: Thank you for your question.  Since MSM contained in food is only a few ppm, MSM intake in this study was mostly derived from MSM supplements. We have obtained food intake data for three days before the measurement date. We did not present dietary survey data because of the poor association between this study and the results of the dietary survey. We added the analysis method of diet survey to p.4 lines 169-171 and described the results in p.7 lines 251-253 (data not shown).

Comment 5: Although the aim of the study is to examine prophylactic effects of MSM, the nature of knee discomfort should be defined more clearly.  Was preceding duration of discomfort or aetiology controlled for e.g. acute versus chronic underlying cause? What score was considered relatively high on the JKOM? Line 108-109?

Response 5: Thank you for your important remarks.  The JKOM total score is an index for evaluating the Quality of Life (QOL) of the entire knee. We rephrased the evaluation of the QOL of the knee as discomfort, but it is unclear as the reviewer said, so we change the notation to "mild knee pain" (p.1 title, p.1 lines 12, p.1 lines 23, p.2 lines 77, p.2 lines 81, p.2 lines 84, p.2 lines 89, p.4 lines 194-195, p.9 lines 267, p.9 lines 276, p.9 lines 286 and p.11 lines 375) . The JKOM score is a questionnaire about chronic symptoms. We added it to the explanation of JKOM (p.4 lines 159-161). We didn't explain enough at Line 108-109. Among the participants who met the participation conditions 1 and 2, we registered in order from the highest overall total JKOM score. In other words, those who met the criteria were enrolled in order of their poor knee QOL. We revised the description (p.3 lines108-109).

Comment 6: Was the absolute decrease in JKOM score in MSM versus placebo groups from week 0 to week 12 significant. 

Response 6: Thank you for your question. The study did not show the amount of change in each score on the JKOM. The amount of change in each item of JKOM from week 0 to week 12 differed significantly only in the item of "health conditions" between the MSM and placebo groups. Add a change table for each item in JKOM to the supplementary materials (Supplementary table 1).

Comment 7: There is discussion of the results and suggestions for future studies that are valid.

Response 7: Thank you for your advice. We added the sentences to the discussion section on mechanisms related to pain (p.9 lines 300-311), antioxidants (p.9 lines 312- p.10 lines 330), inflammatory markers (p.11 lines 331-340), gender-based difference (p.11 lines 341-352) and depression risk (p.11 lines 353-361) in this study.